# Sensory training system for use at home by people with complex regional pain syndrome in England: protocol for a proof-of-concept study

Jessica Coggins  ,[1,2] Sharon Grieve,[1,2] Darren Hart,[3] Alison Llewellyn,[1,4] Mark Palmer,[5] Charlotte Boichat,[6] Candy McCabe[1,4]

[1]School for Health and Social Wellbeing, University of the West of England, Bristol, UK
[2]CRPS Service, Royal United Hospital Bath NHS Trust, Bath, UK
[3]Department of Medical Physics and Bioengineering, Royal United Hospital Bath NHS Trust, Bath, UK
[4]Dorothy House Hospice, Bradford on Avon, UK
[5]Department of Computer Science and Creative Technologies, University of the West of England, Bristol, UK
[6]School of Psychology and Counselling, The Open University, Milton Keynes, UK

**Correspondence to**
Jessica Coggins;
jessica.coggins@uwe.ac.uk

## ABSTRACT

**Introduction** Complex regional pain syndrome (CRPS) is a disabling and distressing chronic pain condition characterised by a range of sensory, motor, autonomic and trophic symptoms. UK guidelines recommend therapy interventions to help normalise touch perception through self-administered tactile and thermal desensitisation activities. Interventions have been developed, aiming to help individuals broaden their sensory experience, thereby relieving chronic pain. However, therapy-led interventions often experience practical constraints and poor adherence. In response, a sensory training system (STS) device has been designed for unsupervised independent home-use.

**Methods** This proof-of-concept study aims to explore whether people with CRPS use the device at home for 30 minutes a day for 30 days. Secondary aims are to determine whether the STS device will change tactile acuity and perceived levels of pain intensity, pain interference, sensitivity or feelings towards the affected limb. We will seek to recruit 20 eligible participants. Participants will be asked to measure tactile acuity using a two-point discrimination assessment, complete an online questionnaire before and after use of the device and complete a daily diary. On completion of the 30-day use, participants will be invited to take part in a semi-structured interview to explore their experiences of using the device.

**Analysis** Pain intensity and pain interference will be scored using the online Assessment Center Scoring Service or using the look-up table in the PROMIS scoring manual. The remaining questionnaire data, including tactile acuity results, and device-use data, including frequency and duration of use, will be analysed using descriptive statistics. Qualitative data will be thematically analysed.

**Ethics and dissemination** London-Stanmore Research Ethics Committee provided a favourable opinion on 19 April 2021 (ref 21/LO/0200). The NHS Health Research Authority, UK, approved this study on 7 June 2021. Dissemination will include peer-reviewed publications, presentations at conferences, social media and reports to the funder and patient charities.

**Trial registration number** ISRCTN89099843.

## STRENGTHS AND LIMITATIONS OF THIS STUDY

⇒ The study team compromises a multidisciplinary team with varying backgrounds.
⇒ Due to the COVID-19 pandemic, the study was re-designed to be delivered remotely, broadening the geographic location of participants.
⇒ Public contributors have been involved in all aspects of the study including design and monitoring.
⇒ Medicines and Healthcare products Regulatory Agency approval gained for the clinical investigation of this non-CE marked sensory training system device.
⇒ Limitations of the study arise from its modest sample size and remote conduct which limits personal contact between researchers and participants; excludes individuals with limited or no access to the online technology; excludes those who are unable to identify a family member or friend to complete the two-point discrimination assessment; and also means the fidelity of this two-point discrimination assessment cannot be guaranteed.

## INTRODUCTION
### Background and rationale

Complex regional pain syndrome (CRPS) is a chronic pain condition which usually develops after trauma to a limb but can occur spontaneously. CRPS often only affects one limb but can spread to involve additional limbs.[1] The condition is characterised by continuing pain disproportionate to the injury, in addition to abnormal sensory, vasomotor, sudomotor and trophic symptoms. The presence of these symptoms is used in the diagnosis of CRPS in accordance with the validated Budapest criteria.[2] One striking feature of CRPS is allodynia; pain due to a normally innocuous stimuli, that is experienced in the affected limb. This can result in increased sensitivity to a range of stimuli, for example, clothing cannot be tolerated over the painful region. The degree of sensitivity may vary between

individuals. People with CRPS often carefully protect their limb to prevent any physical contact to the allodynic site,[3] and they can develop a strong dislike of the CRPS-affected limb.[4] The cause of CRPS is unknown, but evidence suggests that there are multiple mechanisms involved in the pathophysiology.[5 6]

There is limited evidence to estimate the prevalence of CRPS within the UK, however, a study from 2007 suggests that the European incidence rate of CRPS is 26.2 per 100 000 person years.[7] CRPS occurs in approximately 4%–7% of patients who experience limb fractures, injuries or surgery.[6] For many patients with CRPS, symptoms such as swelling, limb discolouration and temperature changes improve within 6–13 months, however, functional limitations persist for some for more than 1 year.[8] Physical impairment and severe pain continue 2 years after initial onset for approximately 15% of patients.[9] These populations experience high levels of disability, and care is primarily focused on reducing pain, preserving or restoring function, and enabling patients to self-manage their condition.[9]

Evidence suggests that patients with CRPS experience a loss of quality of life due to reduced physical health, even more so than patients with other chronic pain conditions.[10] Alongside individual consequences, the annual economic consequences of CRPS are high.[11] This Swiss study found that average treatment costs were 13 times higher, and the number of working days lost 20 times higher within the first 2 years after the accident in people with CRPS compared with those without CRPS.

CRPS is associated with a decreased ability to determine the texture, temperature and location of a stimulus to the painful area.[12] Clinically, the ability to discriminate sensations can be assessed with several methods including a two-point discrimination test. Two probes are applied at the same time to the person's skin, and they are asked to distinguish one or two contact points. The shorter the recorded two-probe distance, the better the sensory discrimination ability. Clinical studies have demonstrated that poor two-point discrimination is correlated with higher intensity of pain experienced and with altered brain representation of the affected body part.[13 14] Interventions have been developed which aim to improve two-point discrimination ability, and therefore reduce pain, by normalising the brain's representation of the painful area. Effectively, these interventions seek to help the individual to broaden their sensory experience, thereby 're-finding' normal sensations and relieving chronic pain.[15]

Sensory discrimination training which involves cognitive effort can improve two-point discrimination, therefore, reducing patient-reported pain. This has been evidenced for electrical stimulation in amputee phantom limb pain (PLP), mechanical stimuli and touch tasks in CRPS, and touch stimuli in chronic low back pain.[15–19] It has been suggested that sensory discrimination training requiring cognitive effort can also have a positive effect on normalising the brain's representation of the painful area.[16] UK

Guidelines for adults with CRPS recommend therapy interventions to help normalise touch perception via self-administered tactile and thermal desensitisation activities.[9] Current sensory discrimination training involves repetitive, manual application of different textiles or stimuli to the 'affected' and an 'unaffected' body site, normally by a therapist or carer.[20 21] However, therapy is slow to deliver improvements, and adherence is poor with limited evidence for any therapist-led interventions that deliver sustained clinically important differences in the long term.[22]

Mindful of the practical constraints of the NHS and poor adherence to sensory discrimination training in practice, a home-based sensory discrimination training intervention has been developed by the research team. It is essential that any device created for this purpose is well-developed, reliable and suitable for unsupervised independent use to gain the expected benefit. A recent study explored the feasibility and acceptability of a novel home-based sensory perception training game for patients with fibromyalgia.[23] Results from this study suggest that a home-based device with a game element is engaging, satisfying and has high adherence. There are commercially available devices that passively stimulate areas of the body but do not include a training element which actively requires cognitive effort in combination with an electrical stimulus. The training element is suggested to be an essential factor in improving two-point discrimination and pain.[17 24]

Our device consists of a commercially available stimulator (RehaMove3, Hasomed, Germany), tablet computer (Surface Go, Microsoft, USA) and wearable arm and leg bands (adapted neoprene sports band, commonly used to hold smartphones) (see figure 1). The bands were adapted for the study to comfortably house an electrode array against the participants' skin. The electrode array comprises four colour-coded electrodes, with five different array sizes to enable the difficulty of the sensory discrimination task to be increased. The electrodes are represented on the tablet screen as four colour-coded images that match the location of the colour-coded electrodes in the array. The participant has the option to play one of two games or use the music player as a relaxation activity.

This current research project builds on 14 years of research and development, throughout which the authors have worked with people with CRPS to design a prototype sensory training system (STS) for use at home to improve tactile acuity.[25–27] The overall goal is to develop a commercially available, clinically effective STS for independent home use by people who have a persistent limb pain, so that they can re-gain normal limb sensation. This protocol describes the clinical testing of the next generation STS that comprises a customised wearable electrode array and game-based training environment, which provides an easy to use and engaging electrical STS for use in the home. The device has received Medicines and Healthcare products Regulatory Agency (MHRA; ref: CI/2021/0019/GB) approval.

**Figure 1** The sensory training system (STS) device.

### Research aim and objectives

This study aims to collect pilot clinical data on a sample of patients with CRPS to determine STS duration of use over a 30-day period and its potential relationship with two-point discrimination and pain intensity.

The primary objective is to explore whether people with CRPS will use the STS device in their own homes adhering to the treatment plan of a minimum of 30 min a day for 30 days (the therapeutic dose). Electrical stimulation of 30 min per day for a 4-week period has been shown in a PLP population to deliver measurable improvements in pain intensity and changes on cortical body maps towards normalisation.[16] The secondary objectives are as follows:

► To determine whether use of the STS device at the therapeutic dose changes tactile acuity (measured using two-point discrimination).
► To determine whether use of the STS device at the therapeutic dose affects perceived levels of pain intensity, pain interference, sensitivity or feelings towards the affected limb.
► To establish whether participants have found using the STS device to be acceptable and feasible in terms of setting up the device, attaching and detaching the electrode array by themselves.
► To establish whether participants have found using the STS device to be sufficiently engaging to encourage adherence to the therapeutic dose.

### METHODOLOGY

This protocol is V.5.0 dated 24 February 2022. Recruitment commenced in September 2021 and is due to close in April 2023.

### Study design

This is a proof-of-concept study designed to explore the acceptability and useability of a customised STS.

### Study setting

In response to the COVID-19 pandemic, this study will be conducted remotely across the UK. This is a single-centre study based at the Royal United Hospitals Bath NHS Foundation Trust (RUH), in collaboration with the University of the West of England (UWE). A collaboration agreement is in place between the RUH and UWE, and any intellectual property is owned by these organisations.

### Participants and sample size

The study aims to recruit 20 adult participants who meet the inclusion criteria and who are willing to use the STS in their homes for 30 min per day for 30 days. This target has been based on prior work by the study team and is a pragmatic sample size based on the scale of funding awarded. Due to the severity of symptoms experienced with CRPS and the common occurrence of 'flares' of symptoms, we nevertheless anticipate a number of withdrawals. All data will be reported, including incomplete data sets and reasons for withdrawal, if provided.

Evidence suggests that the Budapest criteria is the preferred diagnostic tool for adults with CRPS,[2 28] therefore, adults meeting this criteria will be recruited from the CRPS UK Registry (www.crpsnetworkuk.org) or the national CRPS service at the RUH. The registry administrator will email a study invitation letter and the participant information sheet (PIS) to potential participants. If the recruitment target is not met via the CRPS UK

## Box 1 Inclusion and exclusion criteria

Inclusion criteria
⇒ Adults (18+) meeting the Budapest clinical criteria[2] for upper or lower limb CRPS type I.
⇒ An area on their limb where a wearable band with electrode array can be attached above their painful site.
⇒ An average pain level in the last 7 days days rated as ≥5 at rest on a 0–10 scale.
⇒ Access to appropriate technology and willingness to use this technology that enables full study participation (a smart phone, tablet computer or computer compatible with using the video call function on Microsoft Teams, internet access, an email address, and physical and mental capacity to tolerate use of technology).
⇒ A person within their household, carer or a friend to carry out the two-point discrimination measurement.

Exclusion criteria
⇒ Diagnosis of any other neurological, motor disorder or major nerve damage (including CRPS type II).
⇒ Any mental health condition which may detrimentally impede study participation, in the judgement of the patient or researcher.
⇒ The presence of any other limb pathology or pain on the affected CRPS limb.
⇒ Poor skin condition on the area to be stimulated.
⇒ Poorly controlled epilepsy.
⇒ Receiving intensive CRPS-specific multidisciplinary team rehabilitation in an inpatient setting during their participation in the study or within the previous month.
⇒ Unable to understand written or verbal English and give informed consent.
⇒ Active medical implants such as cardiac pacemakers or other devices.
⇒ Exposed orthopaedic metal work in the area of electrical stimulation.
⇒ Pregnancy.
⇒ Known allergy to acrylates which are present in some device components.
⇒ Those living alone who have not formed a 'support bubble' with another household (applies only if social distancing measures, as recommended by the government, are still in place, as there would be no-one able to take the two-point discrimination measurement).

CRPS, complex regional pain syndrome.

Registry, patients of the RUH national CRPS specialist service who have previously consented to be contacted regarding research will be emailed the study invitation letter and the PIS by the researchers. Patients attending the CRPS service will also be provided with the study information by a clinician. Potential participants from the CRPS UK Registry who wish to take part will have their contact details passed to the researcher by the registry administrator. Potential participants recruited via the national CRPS service will contact the researcher directly by email if they wish to take part. The inclusion and exclusion criteria are detailed in box 1.

### Recruitment and consent process
Recruitment will commence with convenience sampling; however, purposeful sampling may be required to try to achieve heterogeneity within the CRPS sample population in terms of range of reported pain intensity, participants

with upper and with lower limb CRPS, and range of CRPS duration.

Once potential participants have expressed an interest after receiving the invitation and PIS, a telephone call will be organised by the research team (JC; SG) in which they will explain the details of the study, answer any questions, confirm eligibility and participant contact details, and explain how to provide online consent using the online consent form, provided using Qualtrics.[29] All participants will be made aware that they are free to decline participation and that doing so will not affect their usual care. If participants would like to continue, a link to the Qualtrics consent form will be emailed to the participant. The consent form will include a copy of the full PIS. Participants will be able to download a PDF copy of the completed consent form for their own records.

Participants may withdraw from the study at any point if they wish. On withdrawal, they will be invited to provide a reason if they choose to help inform future development of the device. It will be made clear in the consent process that any data already collected will be included in the study. Study withdrawal will not affect their continuing care. It is possible that some participants may experience local irritation at the point at which the electrodes are applied to the skin, if this continues beyond an hour after use and causes discomfort, the participant will be advised to withdraw from the study. For minor, earlier indications of skin irritation, the usage and skin advice will be reviewed, with the aim that it is resolved without becoming an issue.

### Intervention design
On receipt of completed consent, the STS will be posted to the participant using a secure courier service and a video-call will be arranged. During the video-call, the researcher will discuss the two-point discrimination assessment with the participant. Participants will also be guided through how to access two films: a short film about the two-point discrimination assessment, and an STS instructions for use video which will show them how to unpack and set up the device and start the game-based training. Both films will be available to access through a private YouTube link. Participants will also have access to paper instructions for the use of the device and how to conduct the two-point discrimination measurements.

Participants will be asked to use the STS device every day for a minimum of 30 min a day. This can be during a single session or divided across multiple sessions over the day as determined by patient preference. To prevent concentration fatigue, it will be recommended that participants do not exceed 2 hours' use in any 1 day. To provide support to participants, they will each be contacted weekly by a member of the research team, either by email or telephone, depending on the participant's preference.

### The STS device
The STS device (figure 2) compromises a pattern of four electrodes placed close to, but not on, the area of

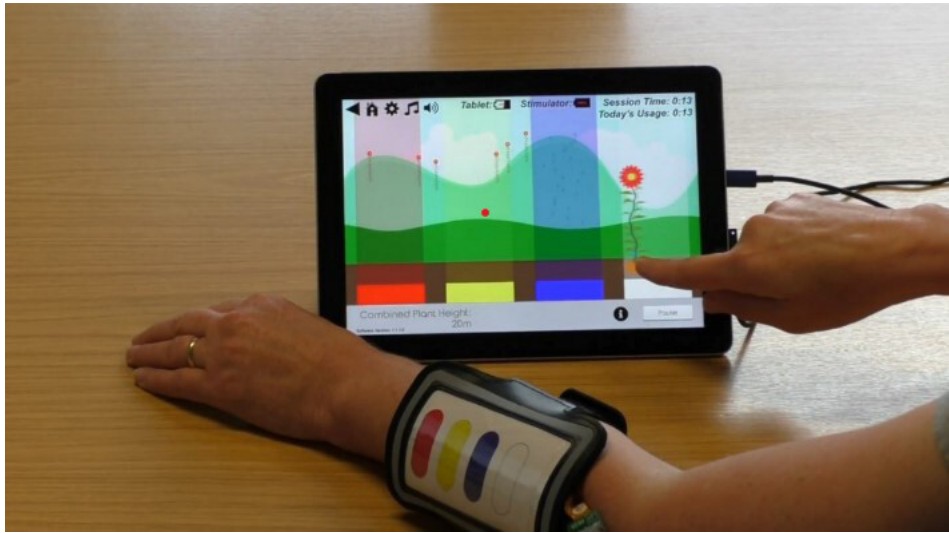

**Figure 2** The STS device in use by a member of the research team. STS, sensory training system.

allodynia using a flexible wearable band. This positioning ensures that the intensity of stimulation is tolerable to the patient and will be less likely to evoke or exacerbate their pain. The electrode array is attached to a stimulator which connects to a tablet computer. The participant uses the computer tablet to set the level of stimulation intensity that each electrodes emits, to the point they can feel the sensation, but it is not painful. Intensity can be controlled within set increments of pulse width between 0% (50 µs) and 100% (350 µs). The participant can set the current between 1 and 50 mA and the frequency at either 10 Hz (slow), 25 Hz (medium) or 40 Hz (fast) depending on their preference. The burst duration of the spatial stimulation will be between 900 and 925 ms dependent on the frequency setting. It is anticipated that the participants will be able to attach the electrode array themselves and they will be able to set the intensity of the stimulation when they first use the device and adjust this at the start of each session if required.

The tablet computer provides a game-based training environment which aims to be easy to use and engaging for participants. The STS application has a choice of two training games (Blockbuster and Flower Tower) and one relaxation activity (Music Player) (see figure 3). Participants can select which training game they wish to use and/or in which order. Each electrode is a different colour and participants are asked to respond by selecting the corresponding colour on the tablet computer. Flower Tower will ask the participant to accurately recognise which electrode is being stimulated. Blockbuster includes five separate levels, participants will be asked to identify which electrode is being stimulated and in more challenging levels, accurately recognise different patterns of stimulation across the array. Participants can repeat the stimulation burst if they wish. Difficulty in the tasks is provided by the stimulation of more than one electrode at any one time, stimulating electrodes that are located adjacent to each other or complexity of the pattern of stimulation. Encouraging feedback messages are given by the tablet computer throughout game play. During the relaxation activity, participants can listen to music and experience the beat of the music as a sensation through the electrodes. The Music Player involves less interaction and therefore will only be accessible after 15 min of game play.

Each participant will receive five different sized electrode arrays and will be able to choose which size they wish to use at the start of training after a discussion with the researcher. The choice will be based on personal preference and practicalities of anatomy. The smaller the electrode array, the closer the electrodes are to each other and therefore, the more challenging the sensory

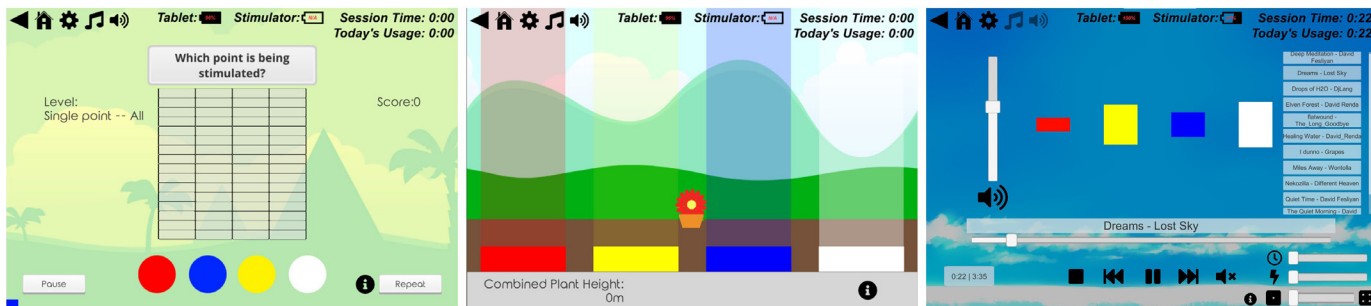

**Figure 3** The STS games and music player. From left to right; Blockbuster single Point all, Flower Tower and the music player. STS, sensory training system.

discrimination task. Participants will be able to progress through from larger to smaller templates if they are scoring well as indicated by the in-play game feedback. If the participant achieves between 50% and 70% accuracy, the device will suggest that this is an appropriate level and that they are using an appropriately sized electrode array. If participants achieve above 70% on multiple occasions, the feedback message will encourage them to consider trying a smaller electrode or moving to the next level. Participants will be asked to record which electrode size is used within a daily participant diary. Once the participant has used the device for 30 days, the device will be collected and returned by a courier at the study's cost. The training protocols are proprietary and devised by UWE and the RUH.

### Data collection and analysis

Participants will be asked to complete a Time 1 Questionnaire 48 hours prior to the planned video call. The questionnaire will be hosted by Qualtrics and will include date of birth, gender, which limb(s) is/are affected, duration of CRPS and current medications. The questionnaire will also include five assessments:

1. Assessment of pain intensity: A self-report rating of average pain intensity over the past 7 days will be given by participants on an 11-point numerical rating scale (0–10) using the Patient-Reported Outcomes Measurement Information System (PROMIS) Numeric Rating Scale V.1.0—Pain Intensity 1a.

2. Assessment of pain interference: A self-report rating of pain interference over the past 7 days will be given by participants on an 11-point numerical rating scale (0–10). This comprises four items in the PROMIS Item Bank V.1.1–Pain Interference–Short Form 4a.[30]

3. Assessment of sensitivity: A self-report rating of average sensitivity over the past 7 days will be given by participants on an 11-point numerical rating scale (0–10). In clinical practice, this patient population would commonly be asked about the degree of sensitivity they are experiencing in their affected limb. Patients' understanding of this would be how tolerant they are to touch and/or temperature changes within that limb. They would be very familiar with this type of question. Our patient research partners were consulted about all study documentation and approved the wording of this question.

4. Assessment of emotional feeling about their affected limb: A self-report rating ranging from strongly positive to strongly negative will be given by participants on an 11-point numerical rating scale (0–10). This is informed by the Bath CRPS Body Perception Disturbance scale.[31]

5. The result of the sensory discrimination assessment will also be entered by the participant on to the Qualtrics survey at time 1. Based on prior work,[14] this assessment measures the ability to discriminate sensation at two points. The smallest distance the participant can correctly identify on three occasions out of two trials,

whether they were touched with one or two points, will be the recorded measurement. On recruitment to the study, participants will be required to identify a friend or family member to conduct this measurement. The participant and their nominated family member/ friend will be guided through this with a short film created by the research team and by an information booklet. The participant will be positioned comfortably with the area to be tested exposed and their eyes closed. A two-point discrimination threshold test using a plastic two-point aesthesiometer will be carried out close to the area of allodynia. Each tool has five different coloured marks, each 2 cm apart, which determine when the aesthesiometer is at 1 cm, 3 cm, 5 cm, 7 cm and 9 cm (figure 4). The person conducting the measurement starts at 9 cm and reduces this according to whether the participant can accurately discriminate between one and two points against their skin. This use of colour codes was designed to simplify the two-point discrimination task for the friend or family member and to reduce the possibility of measurement errors. Ideally, this assessment would have been conducted by a member of the research team but due to COVID-19 requires changes in the study, this pragmatic solution was created. The individual conducting the assessment is also asked to note the anatomical location of the assessment so this can be used again at time 2.

At the end of the 30 days device use, the five assessments will be repeated. Participants will be sent another link via email to complete the Time 2 Questionnaire online using Qualtrics. There will be four additional assessment questions which will measure the overall perception of change of pain intensity, pain interference, sensitivity and emotional feelings towards the affected limb. All will be measured on a seven-point Likert scale ranging from 'very much improved' to 'very much worse'. The sensory discrimination test will again be carried out by a family member or friend and entered into the final section of the Qualtrics survey by the participant.

The questionnaire data will be exported from Qualtrics into Microsoft Excel for the purposes of analysis. Associations between use of the STS device, changes in participants' sensory discrimination, perceived pain intensity, pain interference, sensitivity, emotional feelings about their affected limb and self-report questionnaire data will be investigated. PROMIS interference items will be scored using the online 'Assessment Center Scoring Service' for assessment of fully anonymised data or using the raw score look-up table in the PROMIS scoring manual (https://www.healthmeasures.net/). This service is provided under the auspices of the USA National Institute for Health. Results will be conservatively interpreted in light of the small sample size in this proof-of-concept study.

The primary research aim of investigating whether participants will adhere to using the STS device for the therapeutic dose of 30 min a day for 30 days will be addressed by examining data captured by the device. The

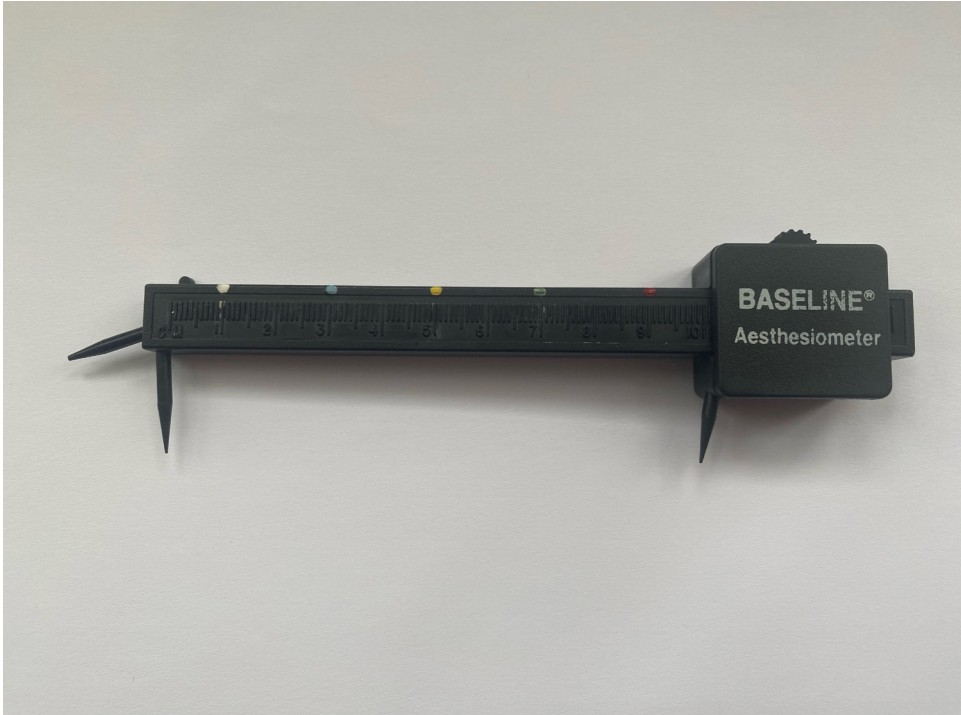

**Figure 4** The marked two-point discrimination tool.

frequency and duration of active game use for each participant will be recorded by the device, and the research team will be able to view this data once the device has been mailed back to the research team.

Participants will be provided with a simple paper diary in which to record changes in array size throughout the 30 days. This will provide an option for them to record any comments on their experience.

Within 2 weeks of each participant completing the 30 days STS device use at home, the researcher will conduct a telephone interview to ask them about their experiences of participating in the study and of using the device. These qualitative data will be recorded with the participant's consent so that it can be transcribed verbatim. The interview schedule will be semi-structured with open-ended questions and will aim to generate qualitative data about the usability and acceptability of the device. Participants will be informed they do not have to answer any questions they do not wish to. Qualitative data from interviews with participants will be analysed using thematic analysis[32] utilising QSR International NVivo software (2018). The summarised study design can be found in figure 5.

### Adverse events

Any adverse events (AEs) will be reported to the chief investigator and documented within the participant's study documentation. If necessary, the AE will be discussed with the research team to decide on the most appropriate action. In the unlikely event of a serious adverse event (SAE) or serious adverse reaction, the study team will immediately inform the chief investigator, sponsor and the MHRA. Any SAEs that are related to the study and unexpected will be reported to the Research

Ethics Committee within 15 days of the chief investigator becoming aware of the event (as per the Health Research Authority guidance).

### Data management

Participant identifiable information, including contact details, will only be seen by the study team. This information will only be used for the purposes of conducting the online surveys and facilitating study contact.

The e-consent and e-questionnaires will be hosted on the Qualtrics as provided and licensed by the UWE, UK. Data on the tablets, which record the participants' device use, will be transferred from the tablet by the study team. The data will be deleted after each individual's participation in the study is complete. Patient confidentiality will be maintained, and data will be collected and retained in accordance with the Data Protection Act, 2018.[33]

Paper-based data will be kept securely in locked cabinets at the RUH or UWE and only accessible to the research team. All electronic personal data will be stored on RUH/UWE password-protected computer systems and managed by the research team. At the end of the study, these documents will be destroyed within 6 months.

Anonymised research data in the study database will be stored at UWE or RUH until the study has been analysed and written for publication. All electronic data will be destroyed within 12 months of study end. Only authorised members of the study team will have access to any electronic data. Data may also be examined by regulatory authorities to check that the study is being carried out correctly. Following the creation of the audio recording from the telephone interview, the file will be password protected and uploaded to a UWE Bristol approved and

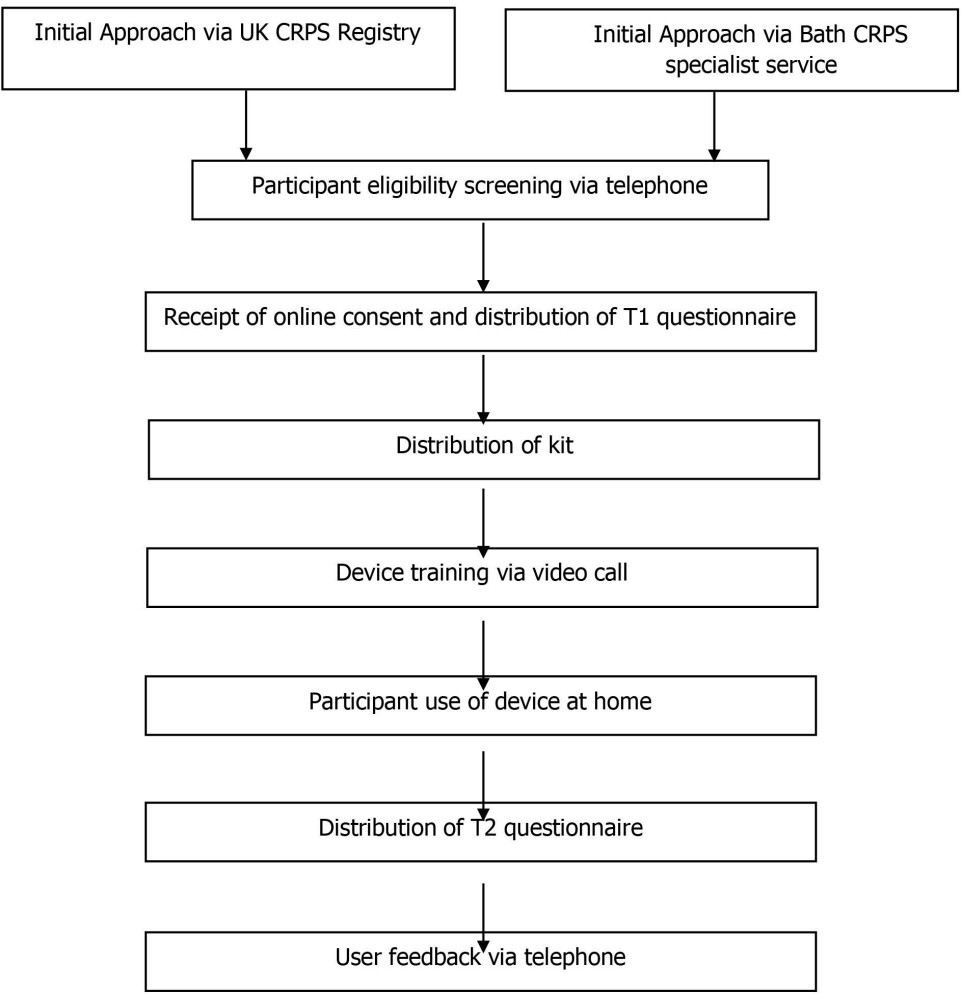

**Figure 5** The study design. CRPS, complex regional pain Syndrome.

recommended transcription service. Audio recordings from telephone interviews will be password protected. Once transcribed, the audio recording will be deleted. The transcriptions will be stored in the same way as other anonymised research data.

### Study monitoring
The Research and Development office at the RUH monitors projects which they sponsor on an annual basis. At UWE, a Research Governance record is audited quarterly. In addition to a UWE ethics application, UWE also require a study risk assessment to be submitted. This is held by UWE.

### Patient and public involvement
Patient and public involvement (PPI) has been an integral part of the project at every stage. It was the patients' desire for a more effective and efficient means of improving sensation in their painful limbs which inspired the original idea and design of the research. Patients have continued to shape the direction of our project and the design of the device through its different developmental stages. Two focus groups have been held and patients strongly endorsed the importance of this device, emphasising the considerable impact CRPS has on people's lives

and the need for new therapeutic approaches. Public Contributors have provided critical feedback to the study design, including recruitment and data capture processes.

Public contributors will be invited to join the Project Management Group (PMG) and will collaborate with the study team to design any patient-related documents. They will also assist in interpretation of findings and dissemination of results arising from the research. Public contributors will contribute to the preparation of future funding applications as co-designing the device with patients has been central to this work from the outset.

### ETHICAL APPROVAL AND DISSEMINATION
This study has been reviewed and given a favourable ethical opinion by London-Stanmore Research Ethics Committee, UK (ref 21/LO/0200) on 19 April 2021 and approved by the NHS Health Research Authority, UK on 7 June 2021. The Medicines and Healthcare Products Regulatory Agency provided a no grounds for objection decision on 4 June 2021. If an amendment to the study protocol is required, the chief investigator will ensure that a valid notice of amendment is submitted, and approval obtained.

On completion of the study, a report will be prepared which will be submitted to the Research Ethics Committee and the funders, vs Arthritis (Award Ref:22029). Dissemination will include peer-reviewed publications, presentations at national/international conferences, social media and reports for relevant patient charities (CRPS UK and Burning Nights). The funder will be acknowledged in the publications and presentations. At the end of the study, the results will be available on the CRPS UK Clinical and Research Network website, and a written summary will be available on request.

Study status: Recruitment and data collection is currently underway and will be completed in April 2023. Data analysis will take place from April 2023 until May 2023. The study is schedule to end in June 2023.

**Acknowledgements** We are very grateful to Evey Watson and Krzysztof Chwiolkam, Games Technology students at UWE, who worked with MP to design the STS application used within this study; and to the CRPS UK registry who will be involved in the recruitment of participants.

**Contributors** All authors qualify for authorship in accordance with the definition provided by the International Committee of Medical Journal Editors (ICMJE). CM, DH, AL and SG conceptualised and designed the study and were involved with funding acquisition, making substantial contributions. DH and MP developed the software. CB and SG initially drafted the protocol, and all authors were subsequently involved with revising and agreeing the final version. JC wrote the first draft of the manuscript based on the study protocol. All authors approved the final manuscript before submission.

**Funding** This work is supported by Versus Arthritis grant number 22029.

**Competing interests** None declared.

**Patient and public involvement** Patients and/or the public were involved in the design, or conduct, or reporting, or dissemination plans of this research. Refer to the Methods section for further details.

**Patient consent for publication** Not applicable.

**Provenance and peer review** Not commissioned; externally peer reviewed.

**ORCID iD**
Jessica Coggins http://orcid.org/0000-0002-6663-6816

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
