## [Reviewer comments · BMJ Open]

ARTICLE DETAILS

TITLE (PROVISIONAL)	A Sensory Training System (STS) for use at home by people with Complex Regional Pain Syndrome in England: protocol for a proof-of-concept study
AUTHORS	Coggins, Jessica; Grieve, Sharon; Hart, Darren; Llewellyn, Alison; Palmer, Mark; Boichat, Charlotte; McCabe, Candy

VERSION 1 – REVIEW

REVIEWER	Christophe Demoulin Liege University
REVIEW RETURNED	28-Dec-2022

GENERAL COMMENTS	This well-written manuscript describes a protocol for a very interesting and original proof-of-concept study. Indeed, it is about a sensory training system for use at home by people with CRPS. However, some important information is lacking at the moment. Please find below some comments to help you to improve the manuscript. Major comments: - Title: the title mention “people with persistent limb pain” whereas the study is about CRPS. Therefore, I would suggest to change the title- Abstract: analysis: very strange that the pre- vs post-test comparison is not mentioned here although the aim of the study is to analyse the changes after 30 days of use.- Introduction: the structure of this section should be improved, some relevant information should be added and recent papers about the topic are not cited:o Very strange to mention only the pharmacotherapy when discussing the treatment of CRPS....o Cite: https://onlinelibrary.wiley.com/doi/10.1002/ejp.1362o Not sure this second paragraph is necessary; however, adding a paragraph about specific consequences,... of CRPS would be relevanto The introduction should include the definition and diagnostic criteria for CRPSo Cite: https://pubmed.ncbi.nlm.nih.gov/34165690/o Regarding the assessment of discrimination, authors should mention that the « two-point discrimination test » is not the only test that has been developed for testing somato-sensory discriminationo Cite: https://www.researchgate.net/publication/344772347_Sensory_discrimination_training_for_adults_with_chronic_musculoskeletal_pain_a_systematic_reviewo Line 35 : a recent study describes a system that available device that encourages independent use to improve sensory discrimination
---

	within the home. This study should be mentioned in your manuscript: https://www.liebertpub.com/doi/abs/10.1089/g4h.2022.0079  - Methods: regarding the inclusion/exclusion criteria, the authors state that “Receiving intensive CRPS specific multidisciplinary team rehabilitation in an inpatient setting during their participation in the study or within the previous month ». Why didn’t you also exclude recent start/changes of a new treatment (drug medication, physiotherapy,...) - Methods: regarding the inclusion/exclusion criteria, will patients with “normal” two-point discrimination be included? - Methods: an important potential issue is related to the two-point discrimination measurement: has the good validity and reliability of this test, when performed at home by non clinicians/investigators been reported ? If it has, please provide the reference. Regarding what is mentioned page 11 (lines 36-50), information is lacking: will the patient be blinded? What measurement tool will be used to do the test? The test will be conducted “close to the area of allodynia” ...what does it mean? Where will it be exactly as the precise location needs to be reliable if you want a reliable measurement... Finally, page 12, line 3: will the retest be performed by the same person ? If not, has your test a good inter-rater reliability when performed by non-clinicians ? - Methods: regarding the location of the electrodes, it is stated page 10, line 17: “our electrodes placed close to, but not on, the area of allodynia »; how is this area of allodynia defined ? - Methods: is it possible to provide some information about the electrostimulation parameters used? Intensity, frequency, impulse length ? - Methods: data collection: several information are lacking: Assessment of sensitivity: has this scale been “validated”? has it an acceptable reliability ? Furthermore, what kind of “sensitivity” is assessed ? ...As there can be several kind of sensitivity: to touch, light, sound, odors,... Minor comments:  - Abstract: the methods to measure tactile acuity should be mentioned - Abstract: the PROMs used in the questionnaire (i.e.PROMIS items) should be mentioned before it appears in the analysis section - Abstract: line 16: “unsupervised” is mentioned. Can you confirm me that your system/software does not allow the investigators/clinicians to follow how the patient are practising everyday ? To know if the exercises has been done, do the investigators/clinicians need to have the system back from the patient ? - Abstract: line 23: it is not clear what you mean by “sensitivity” - Abstract: line 34: “device-use data” are mentioned...but it is not clear what kind of data. - Introduction: line 8: you mention that chronic pain affects around half of the adult population.... But your reference 1 states “affects between one-third and one-half of the population of the UK” - Introduction: page 5, line 39: what do you mean by “training element” ? It is not clear for the readers - Introduction: page 5, line 47: it seems as if the ultimate goal of your system is to improve the two-point discrimination. I suppose it is not the case; it is rather to improve the somatosensory discrimination of patients and thereby, hopefully, decrease their pain/disability - Methods: cite the following reference when mentioning the Budapest criteria: https://pubmed.ncbi.nlm.nih.gov/33230004/ - Methods: it might be relevant to explain why the following exclusion criteria was used “ Known allergy to acrylates”
--	--

	 - Methods: page 9, line 7: it is not clear how heterogeneous your sample should be; how will you determine it is heterogeneous enough ? - Methods: will the results of the game be collected by your system to enable to identify people who performs well and the others? - Methods: as patients will play games, is there an assessment of how funny/enjoyable patients find the games? - Methods: regarding the adverse events, how will they be collected? By a question at the end of the 30-day training ?
--	---

REVIEWER	Max Ortiz-Catalan Chalmers University of Technology, Electrical Engineering
REVIEW RETURNED	31-Jan-2023

GENERAL COMMENTS	The article is well written and provide a general overview of the clinical study. However, some details are lacking to able to ensure reproducibility and adherence to a pre-specified therapeutic protocol. In particular, details on the treatment protocol are lacking. A major limitation of this study is that no follow-up to assess pain is planned, this must be done to test the clinical relevance of the therapy, preferably after several months. I'm guessing that an argument for not performing a follow-up assessment on pain is that the aim of the study is to verify if patient will use the device, rather than the effectiveness of the therapy per se. If this is the argument, please make sure to state that claims on effectiveness will be limited once the results are published, precisely because effectiveness was not the aim of the study. Other comments: The title should reflect that this is a study on CRPS. A limitation that should be noted in the article summary "strengths and limitations" should be the low number of participants. It seems like the claim for which reference 1 is used, only applies for the UK, not the general population. The research aims and objectives are not well matched with the tools to measure them. It says in several places that the aim of the study is to see if the patient will use the device, but this is not stated as an objective. What's the difference between the aim and objectives? None of the objectives seems to correlate with the main aim of device usage. The "data collection and analysis" section doesn't necessarily match with the objectives. It is not clear how the decision of N=20 was made. No power analysis was presented. Please provide an illustration of the electrode placement/arrangement so it can be understood their location with respect the affected area, and the overall setup that the patient needs to have to conduct the therapy. Please also provide a diagram of the treatment session to illustrate the active stimulation time and relaxation time in the form of blocks. It would also be helpful to illustrate the discrimination tasks the patient is expected to accomplish. What does "scoring well" means? Is there a pre-specified range of values? This is with regards the progression from easier to harder in the electrode sets. Please provide the full details of the software Qualtrics and PROMIS. Please provide more details of the assessment of sensitivity. Is this a validated measure? If so, please provide a reference. Please provide the protocol of the 2-point discrimination test to be used or refer to the published one you will use. What would be
--

	orientation of the points with respect to the affected area? Or will it be standardized by anatomical landmarks? Is not clear what information will be logged by the STS device. It is stated that “The frequency and duration of use for each participant will be recorded by the device.” Is this the time the device is on? The software is open? The active stimulation time? The authors declare no competing interest but then it’s not clear who is providing the medical device and the relation of the manufacturer to the study. Was the device specially design for this study? Or is it commercially available? It seems to be approved but no company information was provided. Please provide such information or clarify.
--	---

VERSION 1 – AUTHOR RESPONSE

Reviewer comments	Change to submission	Page number
Title: the title mention “people with persistent limb pain” whereas the study is about CRPS. Therefore, I would suggest to change the title	Thank you for this suggestion, the title has been amended accordingly.	1
Abstract: analysis: very strange that the pre vs post-test comparison is not mentioned here although the aim of the study is to analyse the changes after 30 days of use.	The primary aim is to determine whether individuals with CRPS will use the device in their own homes for 30 minutes a day for 30-days. The pre vs post-test comparison is included in the questionnaire data. The analysis section has been reworded slightly to hopefully provide more clarity.	2
Introduction: the structure of this section should be improved, some relevant information should be added and recent papers about the topic are not cited:	We have re-structured the introduction and included further references.	4
Very strange to mention only the pharmacotherapy when discussing the treatment of CRPS.... Citation	We have removed the focus on pharmacotherapy to avoid confusion. Thank you for this suggestion, we initially included the reference to the UK guidelines for diagnosis, referral, and management in primary and secondary care. As the population for this study is CRPS patients in England, we felt the UK guidelines are more appropriate than the wider European standards.	4
Not sure this second paragraph is necessary; however, adding a paragraph about specific consequences,... of CRPS would be relevant	We have amendment this paragraph to include impact on quality of life and the economic consequences.	4
The introduction should include the definition and diagnostic criteria for CRPS Citation	Thank you for the citation suggestion which we have included alongside the publication reporting the original validation of the Budapest Criteria.	4

Regarding the assessment of discrimination, authors should mention that the « two-point discrimination test » is not the only test that has been developed for testing somato-sensory discrimination Citation	Thank you for highlighting this paper, however, as it reports a systematic review of sensory discrimination training for adults with chronic musculoskeletal pain rather than assessment methods, we do not feel it should be included here. We are aware that there other assessment methods that could be used, however, we describe the selected assessment method for this study.	7
Line 35: a recent study describes a system that available device that encourages independent use to improve sensory discrimination within the home. This study should be mentioned in your manuscript: citation	Thank you for bringing this recent pilot study to our attention, this has now been included.	6
Methods: regarding the inclusion/exclusion criteria, the authors state that “Receiving intensive CRPS specific multidisciplinary team rehabilitation in an inpatient setting during their participation in the study or within the previous month ». Why didn't you also exclude recent start/changes of a new treatment (drug medication, physiotherapy,..)	The authors recognise that attending an inpatient programme could potentially be a challenging, emotional, and demanding experience for patients, especially as many patients travel from across England to attend these inpatient settings. We did not want to create any additional pressures for these patients or interfere with their potential recovery time from the sessions provided during their stay. The primary objective of this study was to explore acceptability and useability, of our new device. Our exclusion criteria would be reviewed for any future work exploring clinical effectiveness.	N/A
Methods: regarding the inclusion/exclusion criteria, will patients with “normal” two-point discrimination be included?	Participants will not be excluded from the study based on their two-point discrimination result. Our aim is the explore any changes in results rather than compare them to ‘normal values’. Previous research that our team has conducted has demonstrated that even when this device is used by healthy volunteers, there are measurable improvements in two-point discrimination so we would not exclude participants who perceive sensations ‘normally’, However, because all participants must be experiencing chronic pain in their CRPS affected limb, we think it highly unlikely that sensory perception to touch would	N/A

	fall within normal parameters.	
Methods: an important potential issue is related to the two-point discrimination measurement: has the good validity and reliability of this test, when performed at home by non-clinicians/investigators been reported? If it has, please provide the reference. Regarding what is mentioned page 11 (lines 36-50), information is lacking: will the patient be blinded? What measurement tool will be used to do the test? The test will be conducted “close to the area of allodynia” ...what does it mean? Where will it be exactly as the precise location needs to be reliable if you want a reliable measurement... Finally, page 12, line 3: will the retest be performed by the same person? If not, has your test a good inter-rater reliability when performed by non-clinicians?	This study was adapted for remote delivery in response to the COVID-19 pandemic, therefore the research team adopted a pragmatic approach to two-point discrimination assessment. The research team have used five different measurement points on the tool, each are 2cm apart and are indicated by 5 different colours. Participants will be asked which colour they successfully reach. This has been included in the manuscript. The participant has their eyes closed during the assessment, however, they will input the result into the questionnaire. We have added further detail about the tool in the data collection section. Participants are encouraged to place the two-point discrimination tool as close to their area of allodynia as possible, but still ensuring personal comfort and to make a note of the position for the subsequent test. The person conducting the assessment will be asked to make a note of the anatomical location of the assessment. We realise that this was not included in the original submission and have added a sentence to reflect this. Inter-rater reliability has not been tested but will be considered for future work.	12
Methods: regarding the location of the electrodes, it is stated page 10, line 17: “our electrodes placed close to, but not on, the area of allodynia; how is this area of allodynia defined?”	Participants are encouraged to place the electrode as close to their area of allodynia as possible, but still ensuring personal comfort to encourage use of the device. Patients who are unable to identify an area on their limb where a wearable band can be attached above their painful site,	N/A

	will be excluded.	
Methods: is it possible to provide some information about the electrostimulation parameters used? Intensity, frequency, impulse length?	The current, frequency and stimulation length has been included under 'the STS device'.	10
Methods: data collection: several information are lacking: Assessment of sensitivity: has this scale been "validated"? has it an acceptable reliability? Furthermore, what kind of "sensitivity" is assessed? ...As there can be several kind of sensitivity: to touch, light, sound, odors,...	This single question was designed by the research team and has not been validated. A single question was felt sufficient for this proof-of-concept study as we did not want to overburden participants with lengthy questionnaire outcome measures. Patients with CRPS experience different types of sensitivity, this question was phrased to capture their personal experience.	N/A
Abstract: the methods to measure tactile acuity should be mentioned	This has now been included.	2
Abstract: the PROMs used in the questionnaire (i.e.PROMIS items) should be mentioned before it appears in the analysis section	Due to the word limit, we have re-worded the analysis sections to ensure that it is clear which are the PROMIS items.	2
Abstract: line 16: "unsupervised" is mentioned. Can you confirm me that your system/software does not allow the investigators/clinicians to follow how the patient are practising everyday? To know if the exercises has been done, do the investigators/clinicians need to have the system back from the patient?	The research team will only have access to the frequency and duration of use data once the device has been posted back to the team by the research participants. We are not able to access these data remotely. We have now clarified this in the data collection section due to the word limit in the abstract. The research team will contact the participants weekly, usage will be discussed at this point.	2
Abstract: line 23: it is not clear what you mean by "sensitivity"	Due to the word limit, this is addressed in the main text. In clinical practice, this patient population would commonly be asked about the degree of sensitivity they are experiencing in their affected limb. Patients understanding of this would be how tolerant they are to touch, temperature changes within that limb.	11

	They would be very familiar with this type of question. Our patient research partners were consulted about all our study documentation and approved the wording of this question.	
Abstract: line 34: “device-use data” are mentioned...but it is not clear what kind of data.	We have added that the device data includes frequency and duration of use of the device.	2
Introduction: line 8: you mention that chronic pain affects around half of the adult population.... But your reference 1 states “affects between one-third and one-half of the population of the UK”	Thank you for highlighting this, this reference has been removed to ensure the introduction relates more to CRPS.	N/A
Introduction: page 5, line 39: what do you mean by “training element”? It is not clear for the readers	We describe the training element in more detail in the section ‘the STS device’.	N/A
Introduction: page 5, line 47: it seems as if the ultimate goal of your system is to improve the two-point discrimination. I suppose it is not the case; it is rather to improve the somatosensory discrimination of patients and thereby, hopefully, decrease their pain/disability	Thank you for highlighting this, we understand that this could be misinterpreted and therefore have changed the text for clarity.	6
Methods: cite the following reference when mentioning the Budapest criteria: citation	Thank you for suggesting this review which demonstrates that the Budapest criteria is the preferred diagnostic tool. This has now been included in the methods section.	7
Methods: it might be relevant to explain why the following exclusion criteria was used “Known allergy to acrylates”	We have clarified that the device components contain acrylates. We do not feel that we need to expand further.	8
Methods: page 9, line 7: it is not clear how heterogeneous your sample should be; how will you determine it is heterogeneous enough?	We anticipate that recruitment for this rare condition may be challenging. Convenience sampling will be used in the first instance. If we are fortunate enough to receive expressions of interests from more than our target number, participants	N/A

	will be selected based upon trying to achieve as much achieve heterogeneity as possible. This will be reported.	
Methods: will the results of the game be collected by your system to enable to identify people who performs well and the others?	The device is able to record the results of the games, however, this will only be used for participants to track their progress and for the software to provide encouraging feedback messages. We are not measuring their skills within the games, but will measure the duration of time they spend on each game.	N/A
Methods: as patients will play games, is there an assessment of how funny/enjoyable patients find the games?	The paper diary provides an opportunity to record any comments on their experience. The semi-structured interviews will also ask about their experiences of the study and using the device. This is described in the data collection methods.	N/A
Methods: regarding the adverse events, how will they be collected? By a question at the end of the 30-day training?	The instructions for use document and video clearly state that participants should report undesirable outcomes, malfunctioning of the device, mistakes in using the device, or injury from the use of this device to the research team. The participants are reminded in their initial video call and weekly check-ins that any adverse events should be reported to the Research team.	N/A
Reviewer 2		
The title should reflect that this is a study on CRPS.	Thank you, the title has now been amended.	1
A limitation that should be noted in the article summary "strengths and limitations" should be the low number of participants.	This is a proof-of-concept study which are usually designed to include fewer participants and aims to determine whether a device is proceedable to avoid resources being wasted.	N/A
It seems like the claim for which reference 1 is used, only applies for the UK, not the general population.	This reference has now been removed.	N/A
The research aims and objectives are not well matched with the tools to measure them. It says in several places that the aim of the study is to see if the patient will use the device, but this is not stated as an objective. What's the difference between the aim and	Thank for you this comment, on reflection we have now amended the aims and objectives for clarity.	6

objectives? None of the objectives seems to correlate with the main aim of device usage. The “data collection and analysis” section doesn’t necessarily match with the objectives.		
It is not clear how the decision of N=20 was made. No power analysis was presented.	As this is an initial proof of concept study, rather than determining clinical effectiveness of our device, the study team considered 20 participants would be sufficient to answer the research aim. This project has been funded by the charity Versus Arthritis and it underwent external peer review scrutiny as part of the award process. The sample size was considered appropriate by reviewers of our grant for the purposes of this study. We are not assessing statistical significance of any outcome, so power calculations were not conducted.	N/A
Please provide an illustration of the electrode placement/arrangement so it can be understood their location with respect the affected area, and the overall setup that the patient needs to have to conduct the therapy.	Each participant had their own individual placement of the electrode array, which is described in the section ‘STS device’.	10
Please also provide a diagram of the treatment session to illustrate the active stimulation time and relaxation time in the form of blocks.	This will vary for each participant, as some may take longer to identify which electrode is being stimulated or which pattern of stimulation. The order of stimulation is randomised across the four electrodes, depending on the degree of difficulty in the task. There is no set stimulation and relaxation time and therefore a diagram would not be useful.	N/A
It would also be helpful to illustrate the discrimination tasks the patient is expected to accomplish.	We have submitted a figure which includes a participant playing the game Flower Tower. We are restricted to the amount we can share at this stage due to IP. Please note that the figure order has been changed since the original submission. The original figure 1 is now figure 2.	
What does “scoring well” means? Is there a pre-specified range of values? This is with regards the progression from easier to harder in the electrode sets.	The device will provide encouraging feedback throughout game play. If a participant achieves between 50-70%, the device will suggest that this is an appropriate level. If a participant achieves between 70-90% accuracy, the device	

	will suggest that they consider trying a smaller electrode. This has been included in the STS device section.	
Please provide the full details of the software Qualtrics and PROMIS.	Thank you, we have amended using the citation recommended by the Qualtrics website and included the full reference in the reference list. PROMIS is cited according to the guidance on www.healthmeasures.net .	9
Please provide more details of the assessment of sensitivity. Is this a validated measure? If so, please provide a reference.	This single question was designed by the research team and has not been validated. A single question was felt sufficient for this proof-of-concept study as we did not want to overburden participants with lengthy questionnaire outcome measures. In clinical practice, this patient population would commonly be asked about the degree of sensitivity they are experiencing in their affected limb. Patients understanding of this would be how tolerant they are to touch, temperature changes within that limb. They would be very familiar with this type of question. Our patient research partners were consulted about all study documentation and approved the wording of this question. This has now been added to the text to clarify.	11
Please provide the protocol of the 2-point discrimination test to be used or refer to the published one you will use. What would be orientation of the points with respect to the affected area? Or will it be standardized by anatomical landmarks?	The participants are provided with an instruction booklet and video describing how to perform the assessment. They are further assisted by an initial video call to ensure any questions are answered. Participants are encouraged to place the two-point discrimination tool as close to their area of allodynia as possible, but still ensuring personal comfort and to make a note of the position for the subsequent test. The person conducting the assessment will be asked to make a note of the anatomical location of the assessment.	N/A
Is not clear what information will be logged by the STS device. It is stated that "The	Thank you for highlighting this, we have now amendment this section to state 'The	12

frequency and duration of use for each participant will be recorded by the device.” Is this the time the device is on? The software is open? The active stimulation time?	frequency and duration of active game use for each participant will be recorded by the device’.	
The authors declare no competing interest but then it’s not clear who is providing the medical device and the relation of the manufacturer to the study. Was the device specially design for this study? Or is it commercially available? It seems to be approved but no company information was provided. Please provide such information or clarify.	The device has been developed by the research team for this study. The stimulator and wearable bands are components that were already on the market and were purchased for this study. This has now been included in the introduction. There have not been any financial contributors from manufacturers. The device has received approval from the Medicines and Healthcare products Regulatory Agency, an agency which regulates medical devices within the United Kingdom to ensure safety and regulation.	5

VERSION 2 – REVIEW

REVIEWER	Christophe Demoulin Liege University
REVIEW RETURNED	05-Mar-2023

GENERAL COMMENTS	Congratulations for this new version of your article. I have only a few last minor commets/suggestions:  - "Strengths and limitations of this study": page 3, line 20: it is not clear if this point is a limitation or a strength - Page 4: could you please check the word "dipropionate" - Page 4: change "Clinically, the ability to discriminate sensations is assessed using a two-point discrimination test" into "Clinically, the ability to discriminate sensations can be assessed with several methods including a two-point discrimination test" - Page 6: Primary aim: shouldn't you state " explore whether people with CRPS will use the STS device in their own homes adhering to the treatment plan of MINIMUM 30 minutes a day for 30 days" ? - Page 9 and elsewhere: it is a bit strange to write "will" considering that the recruitment/experiment is nearly finished (April 2023) - Page 12: regarding the sensory discrimination test conducted again, will it be conducted by the same family member or friend ? I do think that the absence of data regarding the validity/reliability of this test performed by relatives is a limitation in the protocol.
--

REVIEWER	Max Ortiz-Catalan Chalmers University of Technology, Electrical Engineering
REVIEW RETURNED	19-Mar-2023

GENERAL COMMENTS	Thank you for addressing some of my comment. However, not all of
--

	them were sufficiently addressed and there remains a considerable lack of transparency on the treatment sessions and conflict of interests. As I wrote before, a limitation of this study is the low number of participants and this must be acknowledged in the “strengths and limitations”. It doesn’t matter that it is a proof-of-concept study, it is still a limitation! The authors didn’t answer my question of how the sample N=20 was selected. That the sample was approved by the funding body says nothing about why N=20 is suitable. You said you considered it appropriate, ok, but what makes it appropriate? That is the question. If it was arbitrarily decided, just say it. If you only got money to do 20, that’s alright too, just be transparent for the reason N=20. I disagree with the author’s comment that a diagram for the training session wouldn’t be useful. I wouldn’t be asking for it, if the treatment sessions would be so clear. The whole point of writing a protocol article is to explain how you will be conducting the intervention, so you should be able to explain that for others to reproduce or understand the reasons why you might get the outcomes you will get. If the time varies use a range of time. Again, please provide a diagram/illustration of the treatment sessions. Similarly as above and regarding your answer about the discrimination tasks, the whole point of writing a protocol article is to publish the treatment protocol! That includes how the treatment is conducted. I suppose you have filed a patent for the system so I don’t see the problem with IP. If not, file a patent and then publish the protocol. If you can’t provide the information on how the treatment will be conducted, there is no point of having this article published. Your response regarding the 2-point discrimination test is insufficient. Good that they have a video and booklet, but again, what protocol for the test are you using? You made it yourselves or is it a published one? If you made it, provide it with the article, and if it’s a published one, provide the reference. The information provided in the answer should be also included in the article. I suppose that the area of the test will remain the same for a given patient throughout all tests, write this in the article as well. Regarding competing interest. The authors said above that they have restrictions on IP but also declare no competing interest, this is seems contradictory. If the stimulator and wearable bands are components on the market, provide the information of the product model and manufacturer in the article. This is common practice. Also, specify in the article where those IP restrictions come from.
--	---

VERSION 2 – AUTHOR RESPONSE

Reviewer comments	Change to submission	Page number
"Strengths and limitations of this study": page 3, line 20: it is not clear if this point is a limitation or a strength	This has been amended to clearly state that this point is considered a limitation, thank you for highlighting this.	3
Page 4: could you please check the word "dipropionate"	Thank you very much for highlighting this, the typo has been corrected.	4

Page 4: change "Clinically, the ability to discriminate sensations is assessed using a two-point discrimination test" into "Clinically, the ability to discriminate sensations can be assessed with several methods including a two-point discrimination test"	Thank you for this suggestion, we have amended this sentence accordingly.	4
Page 6: Primary aim: shouldn't you state " explore whether people with CRPS will use the STS device in their own homes adhering to the treatment plan of MINIMUM 30 minutes a day for 30 days" ?	The text has been amended in line with this suggestion.	6
Page 9 and elsewhere: it is a bit strange to write "will" considering that the recruitment/experiment is nearly finished (April 2023)	Thank you for highlighting this and we appreciate your perspective given that recruitment is ongoing. However, it is customary to present a study protocol in the future tense. Therefore, we have used 'will' throughout the article.	N/A
Page 12: regarding the sensory discrimination test conducted again, will it be conducted by the same family member or friend ? I do think that the absence of data regarding the validity/reliability of this test performed by relatives is a limitation in the protocol.	Yes, it is intended that the same family member or friend will conduct both sensory discrimination tests. For clarity, we have amended the text in the data collection section. We acknowledge that the possibility that the conduct of this test by a family member/friend rather than a health care professional may impact the fidelity of the assessment. This has now been included in the limitation section.	12
Reviewer 2		
As I wrote before, a limitation of this study is the low number of participants and this must me acknowledge in the "strengths and limitations". It doesn't matter that is a proof-of-concept study, it is still a limitation! The authors didn't answer my question of how the sample N=20 was selected. That the sample was approved by the funding body says nothing about why N=20 is suitable. You said you considered it appropriate, ok, but what makes it appropriate? That is the question. If it was arbitrarily decided, just say it. If you only got money to do 20, that's alright too, just be transparent for the reason N=20.	Thank you for this observation, we accept that the number of participants is small, however, it is in line with similar studies in CRPS and reflective of this rare condition. Whilst we have been careful to specify that this a proof-of-concept study not a trial of efficacy, we have, however, included this as a potential limitation. In response to your question, a sample size calculation was not conducted a priori as our study represents exploratory	7

	proof of concept work. Prior work conducted by the study team previously demonstrated the challenges of recruiting people with CRPS to a study of this nature; we were also cognisant of the limits of our funding award and therefore 20 participants was considered a pragmatic and realistic target. We have amended the text accordingly.	
I disagree with the author's comment that a diagram for the training session wouldn't be useful. I wouldn't be asking for it, if the treatment sessions would be so clear. The whole point of writing a protocol article is to explain how you will be conducting the intervention, so you should be able to explain that for others to reproduce or understand the reasons why you might get the outcomes you will get. If the time varies use a range of time. Again, please provide a diagram/illustration of the treatment sessions.	Thank you for reiterating this point. Whilst we would like to have been able to provide a diagram/illustration, the nature of a treatment session is dependent upon the user in that they can select which game they wish to play, which level, in which order, and for how long. Response times within each game are also entirely user dependent. User instructions specify that the device should be used for a minimum of 30 minutes per day for 30 days, but it is at the discretion of the user as to how long they spend in each activity within this time frame. Users are also able to complete either a single session or multiple sessions to equate to a minimum of 30 minutes use within each day. We will incorporate a figure with screenshots of each game, however, without a real time video, it is impossible to fully convey all permutations of the treatment session as experienced by each individual user. We have amended the text for clarity.	10
Similarly, as above and regarding your answer about the discrimination tasks, the whole point of writing a protocol article is to publish the treatment protocol! That includes how the treatment is conducted. I suppose you have filed a patent for the system, so I don't see the problem with IP. If not, file a patent and then publish the protocol. If you can't provide the information on how the treatment will be conducted, there is no point	Thank you for this comment. Our legal team have advised us that the device is not patentable as the hardware components we are using are in the public domain. The software programme is novel, but as we imagine the Reviewer is aware, getting a patent for software is problematic.	N/A

of having this article published.	We have provided a picture of our Sensory Training System and the type of games that each participant can engage in. As already stated in our Introduction text 'The electrode array comprises four colour coded electrodes, with five different array sizes to enable the difficulty of the sensory-discrimination task to be increased. The electrodes are represented on the tablet screen as four colour coded images that match the location of the colour coded electrodes in the array.' We believe this provides sufficient information for the reader to understand how the participant conducts the discrimination task.	
Your response regarding the 2-point discrimination test is insufficient. Good that they have a video and booklet, but again, what protocol for the test are you using? You made it yourselves or is it a published one? If you made it, provide it with the article, and if it's a published one, provide the reference. The information provided in the answer should be also included in the article. I suppose that the area of the test will remain the same for a given patient throughout all tests, write this in the article as well.	Thank you for your acknowledgement about the value of the video and booklet. The protocol for the two-point discrimination test was adapted from prior work already cited in the article. For clarity, we have inserted this citation again in the description of the assessment in the data collection section. Participant instructions are to conduct the sensory discrimination test on the same anatomical location both at baseline and following completion of use of the device. Please see point 5 in the data collection section where we had already described this requirement.	12
Regarding competing interest. The authors aid above that they have restrictions on IP but also declare no competing interest, this is seems contradictory. If the stimulator and wearable bands are components on the market, provide the information of the product model and manufacturer in the article. This is common practice. Also, specify in the article where those IP restrictions come from.	The details of the commercially available components have been included in the article. IP restrictions arise from the contractual agreements between the institutions of the study team members and in line with the funding agreement.	6

VERSION 3 – REVIEW

REVIEWER	Christophe Demoulin Liege University
REVIEW RETURNED	22-Apr-2023

GENERAL COMMENTS	Well done!
------------

REVIEWER	Max Ortiz-Catalan Chalmers University of Technology, Electrical Engineering
REVIEW RETURNED	04-May-2023

GENERAL COMMENTS	There is still not enough information to know what is meant to be trained on the sensory system, is it spatial, temporal, or both? And how? The authors seems to be unable to provide this information because of IP restrictions but also they don't clearly specify the conflict of interest. Then write, the "training protocols are proprietary" and then mention who owns them. Regarding commercially available components, the standard is Model, Company, Country of the company. Please update the information. There is a clear conflict of interest in this work in which the authors are prevented to specify details of the treatment and the system, and these conflicts of interest remain unspecified in the article. You mentioned that "IP restrictions arise from the contractual agreements between the institutions of the study team members and in line with the funding agreement." Ok, this must be specified in the article and also name the actors that hold the IP and who prevent further disclosures. Saying that is in line with the funding agreement is all fine, the readers should still know who stands to gain from this work if commercially successful. You have consulted with your legal team concerning patents, which means there is or has been commercial intention. If you can't get a patent, the other reason for not disclosing all the details is that you want to keep it as an "industrial secret", whoever stands to win financially from such a secret must be mentioned in the article. There is nothing wrong with filing a patent or preserving information "secret" for commercial purposes, this just needs to be disclosed. Who did all the integration work? I suppose that is the party who owns the IP and should be specified.
--

VERSION 3 – AUTHOR RESPONSE

Reviewer 2		
There is still not enough information to know what is meant to be trained on the sensory system, is it spatial, temporal, or both? And how? The authors seems to be unable to provide this information because of IP restrictions but also they don't clearly specify the conflict of interest. Then write, the "training protocols are proprietary" and then mention who owns them.	This is spatial simulation; we have included this within page 10. We have also provided further details around the parameters of pulse width that users have control of. The STS device section explains the burst duration and premise of each game. We have added a sentence to clarify that bursts can be repeated. Authors feel this is sufficient information for readers to understand the Sensory Training System.	

	The training protocols were proprietary and devised by the University of the West of England and Royal United Hospitals Bath NHS Foundation Trust, this has been added to the manuscript.	
Regarding commercially available components, the standard is Model, Company, Country of the company. Please update the information.	Thank you, this has been amended within the manuscript.	
There is a clear conflict of interest in this work in which the authors are prevented to specify details of the treatment and the system, and these conflicts of interest remain unspecified in the article. You mentioned that “IP restrictions arise from the contractual agreements between the institutions of the study team members and in line with the funding agreement.” Ok, this must be specified in the article and also name the actors that hold the IP and who prevent further disclosures. Saying that is in line with the funding agreement is all fine, the readers should still know who stands to gain from this work if commercially successful. You have consulted with your legal team concerning patents, which means there is or has been commercial intention. If you can't get a patent, the other reason for not disclosing all the details is that you want to keep it as an “industrial secret”, whoever stands to win financially from such a secret must be mentioned in the article. There is nothing wrong with filing a patent or preserving information “secret” for commercial purposes, this just needs to be disclosed. Who did all the integration work? I suppose that is the party who owns the IP and should be specified.	There is a collaborative agreement between the University of the West of England and the Royal Untied Hospitals Bath NHS Foundation Trust and resulting intellectual property is owned by these organisations, this has now been included in the manuscript. We disagree that there is a conflict of interest at this stage. Our ultimate goal is to develop a commercially available device, which we hope would provide patient benefit. At this stage of our work, we do not know what that commercial offer may look like, in terms of design of the device or financial package. The aim of this proof-of-concept study is to explore whether CRPS patients find the software and hardware of this prototype device to be sufficiently engaging and functional for them to use it on a regular basis over a 30 day period. If our study data confirms this then we require further investment and research to continue to develop our device to a commercially acceptable design, and then test its clinical effectiveness within a randomised trial.